# In Vivo Analysis of Embryo Development and Behavioral Response of Medaka Fish under Static Magnetic Field Exposures

**DOI:** 10.3390/ijerph16050844

**Published:** 2019-03-08

**Authors:** Weinong Sun, Yaqing He, Sai-Wing Leung, Yuen-Chong Kong

**Affiliations:** 1Department of Electronic Engineering, City University of Hong Kong, 83 Tat Chee Avenue, Hong Kong, China; yaqinghe2-c@my.cityu.edu.hk (Y.H.); eeswl@cityu.edu.hk (S.-W.L.); 2Department of Chemistry, City University of Hong Kong, 83 Tat Chee Avenue, Hong Kong, China; bhrkong@cityu.edu.hk

**Keywords:** static magnetic field, medaka fish, in vivo, embryo development, behavioral response

## Abstract

The static magnetic field (SMF) in human exposure has become a health risk concern, especially with respect to prolonged exposure. The International Commission on Non-Ionizing Radiation Protection (ICNIRP) has been considering cell or animal models to be adopted to estimate the possible human health impacts after such exposure. The medaka fish is a good animal model for human-related health assessment studies; this paper examines both the embryo development and behavioral responses in medaka fish in vivo to long-term SMF exposure at the mT level. SMF exposure was examined for the complete developmental period of embryos until hatched; the embryos were monitored and recorded every 24 h for different morphological abnormalities in their developmental stages. The behavioral response of adult fish was also examined by analyzing their swimming velocities and positioning as compared with that of the control group. It was observed that there were no impacts on embryo development under prolonged exposure up to about 100 mT while the swimming behavior of the adult fish under exposure was different to the control group—the swimming movement of the treated group was more static, with an average velocity of 24.6% less as observed over a 24-h duration.

## 1. Introduction

Static magnetic field (SMF) exposure involving a large DC current generating a high level of magnetic fields is a safety concern. The International Commission on Non-Ionizing Radiation Protection (ICNIRP) has been considering the effect of exposure by using cell or animal models to assess the possible health risk for humans [1]. There are numerous studies on the biological effects of static magnetic fields in vitro and in vivo [2]. It is evident that SMF affects several endpoints at an intensity in the mT range (as shown in several reviews on in vitro studies [3,4,5,6]), elucidating the interaction mechanisms of SMF exposure to cells and DNA, with the same implications in in vivo studies as in health assessment models [7,8,9]. The ICNIRP and WHO have also reviewed the cellular and animal studies of static field exposure, and it was concluded that in vitro studies are not sufficient to identify health effects without corroborating evidence from in vivo studies [10]. There are few studies on in vivo SMF exposure to link human health risks with corroborating evidence; the experimental outcomes are also very limited in terms of providing evidence for conclusive prolonged SMF exposure.

It is not uncommon to record an SMF of up to 2 mT inside the passenger cabin in trains [11,12,13]. Additionally, SMFs of up to 100 mT in certain accessible locations have been measured in production plants for aluminum, with an average of 10 mT in electrolytic processing plants [14]. An SMF of 0.95 mT inside the hybrid e-vehicles has also been reported [15].

There are few studies available on the developmental and the neurobehavioral responses for animal experiments in SMF for prolonged exposure; this paper focuses on providing experimental evidence on in vivo studies using medaka fish for correlation with human responses. The medaka fish has been considered as a good animal model for developmental and behavioral studies including that of the embryo, and the growing fish is especially sensitive to many environmental conditions [16,17,18]. Fish models have been frequently used in environmental risk assessment, as the molecular mechanisms underlying germ cell development in fish are comparable to those in mammals, including humans [19]. The medaka fish, with genome sequences well-annotated, is a typical aquaria fish model for human diseases as a research organism [20,21,22], for which there are established resources for genetic and genomic studies [23]. The role of histones in regulation of key developmental genes has also been described in medaka fish [24]. It is the purpose of this paper to examine the implications of possible biological and behavioral responses on human exposure by in vivo studies using medaka fish.

## 2. Materials and Methods

### 2.1. Embryo Development

#### 2.1.1. Experimental Setup

Marine medaka fish embryos for the experiment were prepared by natural spawning and cultured in the laboratory in artificial seawater at 28.5 °C; embryos at the pre-mid-gastrula stage were selected for the experiment. The embryos were separated into individual culture dishes for observation—five dishes for the treated group with SMF exposure, and three dishes for the control group without exposure. The dishes were 9 cm in diameter and contained 15 fish embryos each. The embryos at the pre-mid-gastrula stage in the culture dish under a microscope are illustrated in Figure 1.

Figure 2 illustrates two coupled parallel disk magnets with a diameter of 10 cm, arranged for the static magnetic field of the experiment; the dishes with the fish embryos were placed in the middle of the two disk magnets. The measurements of the magnetic strength of the embryo exposure region were made by a Hirst GM5 Gauss meter; the field strength in terms of the layers between the magnets is included in Figure 2a. The size of the medaka embryos in all developmental stages is relatively small, and the selected 15 embryos occupied only a small part of the dish. In our setup, the 15 embryos in each dish were placed in the middle of the dish as exposure region with water droplets extruded from a dropper.

Figure 3 illustrates the culture dishes and the magnet arrangement of the exposure. The embryos of the treated groups in culture dishes labeled 1 to 5 as the treated groups were under exposure to the SMF, with exposure durations of 1 day, 3 days, 6 days, 10 days, and 15 days, respectively. The embryos in the culture dishes labeled 6 to 8 as the control groups were under sham exposure. Dishes 2 to 5 were moved to the control group for further sham exposure after 1, 3, 6, and 10 days of exposure for further observation.

The whole arrangement of the embryo dishes with the magnets were placed in a room temperature environment of 23–24 °C, which is compatible with the growth environment of the medaka fish embryos. 

#### 2.1.2. Methodology

The embryos were monitored every 24 h and recorded using a digital camera (SPOT-RT, Diagnostic Instrument Inc., Sterling Heights, MI, USA) connected with a Nikon compound microscope (Nikon Eclipse TE200, Nikon, Tokyo, Japan), for a minimum period of 15 days, or until hatched; their different morphological abnormalities were monitored and recorded. The morphological features for observation included body length, head circumference, eye symmetry, spine shape, tail shape, and heart shape. The morphological abnormality ratio and the death rate of embryos were also recorded. 

The *t*-test was adopted in the results analysis to determine if two sets of data were significantly different from each other; the significance of the embryo developmental stages between the treated and the control groups, among different exposure durations, was analyzed. *p* >|*t*| represents the 2-tailed *p*-values used in testing the null hypothesis that the regression coefficient is 0, where *t* = Coef/Std.Error, which is the regression coefficient divided by standard error. An alpha of 0.05 corresponds to a significant level of 5%, which was adopted in our study; the smaller the *p* >|*t*| the more significant it was [25].

### 2.2. Behavioral Response

#### 2.2.1. Experimental Setup

Only adult male medaka fish grown in normal laboratory environment were adopted as the animal model in this experiment. The fish were maintained under a daily light/dark cycle at room temperature at 23–24 °C during the experiment. There were two groups of fish for the experiments of behavioral response—the treated group was exposed under static magnetic fields, while the fish in the control group were subjected to sham exposure. The movements of both groups were recorded by two separate webcams capable of recording 1280 × 720 videos by two HP Pavilion DV2 Notebooks. The recorded videos were then converted into numerical data by an object positioning and tracking program in terms of the swimming path. The behavioral response of the two groups was analyzed by the swimming velocities and their location distributions of individual fish in the tanks.

Figure 4 illustrates the experimental setup. In order to minimize any light refraction by the water into the tank from the fluorescent lighting of the laboratory, white papers were used to cover the top of the camera, the bottom and periphery of the fish tank. Two Light-emitting diode (LED) desk lamps were also used for an even luminary strength as viewed from the webcams. Figure 5 illustrates the magnet adopted in the treated group, with the North Pole pointing upward, and a screen capture of the display from the video clip by the webcam.

#### 2.2.2. Methodology

An object positioning program was used to recognize the fish. The coordinates of fish per frame were then converted to the video frames of the fish movement in numerical data, by software coding of object detection and mapping algorithms. Frames of the videos of the control group and the treated group were both cropped from 640 × 480 pixels to 380 × 380 pixels to the area of fish tank for editing into 24 clips of videos of 5-min length for analysis. Background model training and noise filtering were adopted, and background subtractions from each frame were also adopted for image segmentation and object detection of the fish; this process is illustrated in Figure 6. The method of object mapping was used to distinguish different individual fish, the objects were mapped by the shortest Euclidean distance between two consecutive frames, by:(1)$$\sqrt{{\left(s_{x_{i}}-s_{x}{}_{{}_{i-1}}\right)}^{2}+{\left(s_{y_{i}}-s_{y}{}_{{}_{i-1}}\right)}^{2}}$$
where *S*_*x*_, represents the *x*-, *y*-coordinate of the fish correspondingly, and *i* represents the number of frames.

The average velocity of fish per video was also calculated. The position distributions of all the fish are displayed by heatmap plotting. The video clips were converted into data files with coordinates by an object positioning program; this is illustrated in Figure 7 as a frame marking with “O” on the coordinates. In each video clip, the total swimming distance in terms of pixel was calculated by the superposition of the shortest Euclidean distance between two consecutive frames in all frames, dividing this total distance by the total time of the video clip, which is 300 s. The average velocity of all fish was hence calculated as:(2)$$\frac{1}{60\times 5}\sum_{i=2}^{n_{c}}\sqrt{{\left(s_{x_{i}}-s_{x_{i-1}}\right)}^{2}+{\left(s_{y_{i}}-s_{y_{i-1}}\right)}^{2}}$$
where *n_c_* represents the total number of frames, and *s_x_*, represents the *x*-, *y*-coordinate of the fish correspondingly.

## 3. Results

Both results of the embryo development and the behavior responses of the medaka fish are presented here.

### 3.1. Embryo Development

Table 1 shows the embryos hatching status in all the dishes for embryo development; the average period needed for embryo hatching was around 15 days for both treated and control groups of the experiment. Embryos that had not hatched after 20 days were abandoned; similar hatchability rates were observed for all the experimental groups. Figure 8 and Figure 9 show the typical stages of embryo development every 24 h in the dishes until hatched, at Stage 40. Stage 40 is the “first fry” stage, with the medaka fish hatching with caudal and pectoral fins [26]. 

Figure 8 illustrates the progress of the developmental stages of the treated group, with 15 days exposure, in comparison with that of the control group. The trend of the two fold lines is very close—the *p*-value of 0.91 by the paired *t*-test indicates that there is no significant difference in their embryo developmental stages between the treated and the control groups. 

Figure 9 shows the progress of the developmental stages of the treated groups with exposure durations of 1 day, 3 days, 6 days, 10 days, and 15 days—all the *p*-values are larger than 0.9 among any two durations, implying no significant difference in the embryos’ developmental stages with different exposure durations. It was observed in the experiment and is noted in Figure 9 that there are two abnormal embryos—one abnormal embryo was detected in dish 5 after 3 days of exposure with full-body morphological abnormalities, and another abnormal embryo was detected in dish 2 after 7 days of exposure with abnormal eye development, showing nonlinear response to the exposure duration.

Figure 10 illustrates the two observed abnormal embryos above, with an abnormal rate of 2.7% of all the embryos being observed in the treated groups. The low abnormal rate can be considered as an occasional event such as accidental bacterial infection or developmental failure within ordinary probability [27,28,29]. Table 2 illustrates the appearance of the normal developed medaka embryos of the experiments at: stages of (a) 14, (b) 18, (c) 22, (d) 23, (e) 28, (f) 29, (g) 30, (h) 33, (i) 35, (j) 37, (k) 39, and (l) 40 of the 15-day treated group and the control groups. Figure 11 illustrates photographs of larvae at 3 and 13 days after hatching.

SMF exposure did not affect the progress of embryo development.1/15 abnormal developed embryos were detected in dish 5 after 3 days of exposure1/15 abnormal developed embryo were detected in dish 2 after 7 days of exposureNo-linear response, hypothesis of occasional event.

### 3.2. Behavioral Response

The behavioral response in terms of the average swimming velocities of individual video clips of different time slots is illustrated in Table 3; Equation (2) was applied for the evaluation of the velocities. 

A paired *t*-test was carried out for the comparison of the velocities of the two groups. It was evidenced that with a *p*-value of 0.02, which is less than 0.05 at a confidence interval of 95%, that the swimming velocities of the two groups were different. The average velocity of the treated group was slower than the control group by 24.6%. 

Figure 12 illustrates the heatmap charts of the treated group and the control group; the heatmap charts are in pixel format, showing the fish dwelling in a particular pixel in terms of the coordination of the fish of all the 12 video clips of each group. Figure 13 shows an overlapping of the magnetic field strength to the heatmap chart of the test group, while there was no influence of the magnet to the control group. It was observed that fish of the treated group were likely to swim at the center, and otherwise for the control group—this can be postulated as that there is a preference of the fish to dwell under the influence of SMF. An alternate explanation is that the fish disliked the higher field strengths in the concentric ring pattern above 100 mT and were trying to avoid this high field strength. The fish would either move to the corner or to the center; either way they would be out of the higher field strengths of the concentric ring pattern. The fish in the treated group were more likely to swim to the light source than the control group.

## 4. Discussion

This paper describes an in vivo experimental study of the marine medaka fish model under static magnetic field exposure; both the induced developmental response of the fish embryo and the behavioral responses of the grown fish are presented. 

The developmental stages of embryos were observed for any abnormalities, and the analysis of variance of the fish swimming velocities was performed for assessing group differences with and without SMF exposures. It was observed that there was no impact on the embryos development under prolonged exposure for the whole developmental period for 15 days under 100 mT. The swimming velocities of the grown fish under exposure were observed to be statistically significantly different to that of the control group, with a *p*-value of less than 0.05. 

As experiments of human responses under the influence of any SMF might be difficult to be carried out; the outcomes of the developmental response and the behavioral response of the medaka fish model can provide a preliminary inference to that of the human model. 

The strength of SMF of the experiments for the medaka fish in this study is larger by several folds as compared to that of the daily environment including that of inside of e-vehicles; however results have concluded that SMFs do have a certain influence on the behavioral responses of the fish. The experiment of this study would serve as an implication that the influence of SMF under prolonged exposure may warrant further studies for a human model, especially for e-vehicle applications for passengers and drivers.

## 5. Conclusions

The outcomes from the in vivo experiments suggest that prolonged exposure of medaka embryos to SMF has no impact on embryo development, and no obvious higher probability of embryo malformation or differences of growth cycle was observed in the experiments for 15 days. It was also observed that under prolonged exposure of grown medaka fish, the swimming behavior, average swimming velocity, and swimming positions were different between the treated group and the control group, and that there was a preference of the fish to dwell under the influence of SMF.

## Figures and Tables

**Figure 1 ijerph-16-00844-f001:**
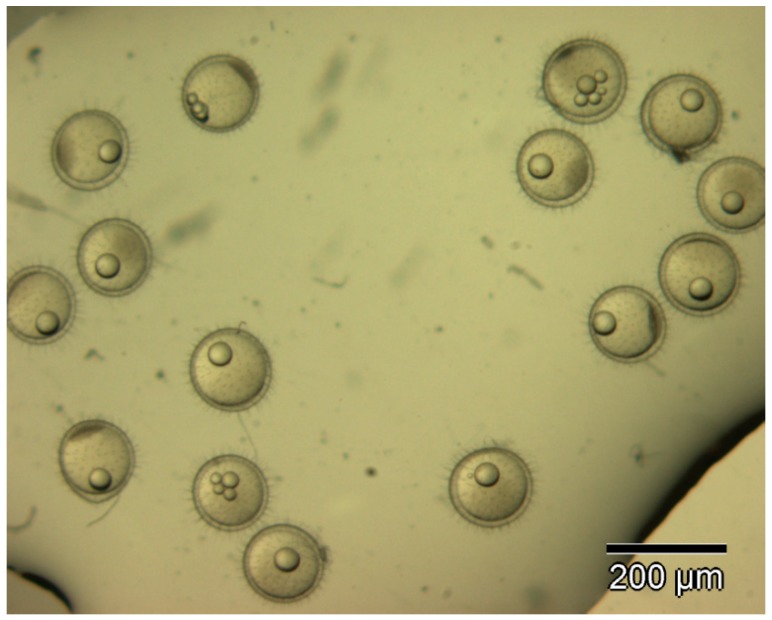
Medaka embryos in culture dish under microscope.

**Figure 2 ijerph-16-00844-f002:**
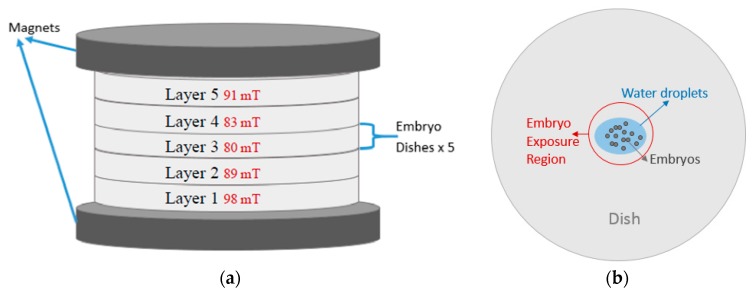
Magnet configuration for treated groups: (**a**) side view; (**b**) upper view.

**Figure 3 ijerph-16-00844-f003:**
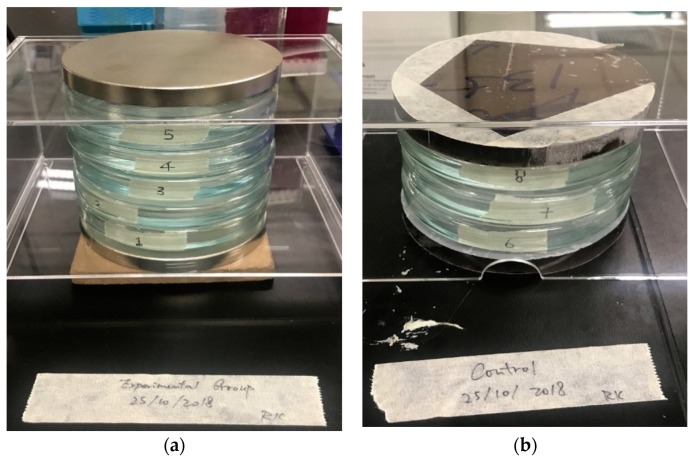
Exposure arrangement of: (**a**) treated groups; (**b**) control groups.

**Figure 4 ijerph-16-00844-f004:**
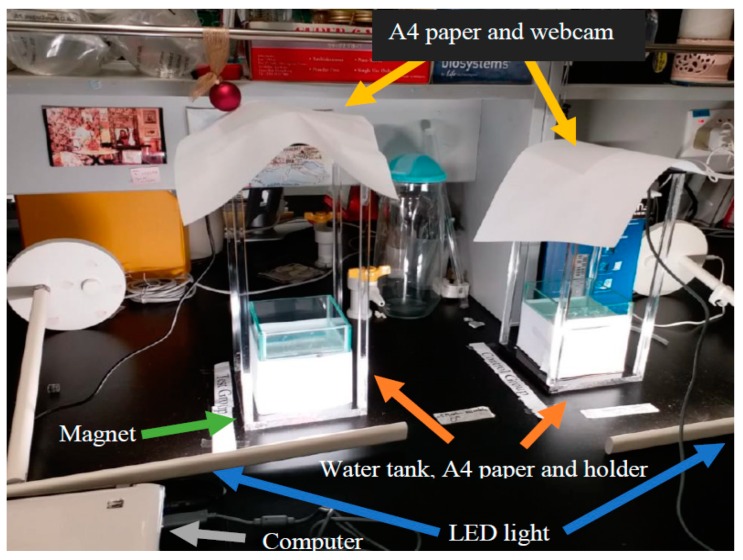
The setup of the experiment.

**Figure 5 ijerph-16-00844-f005:**
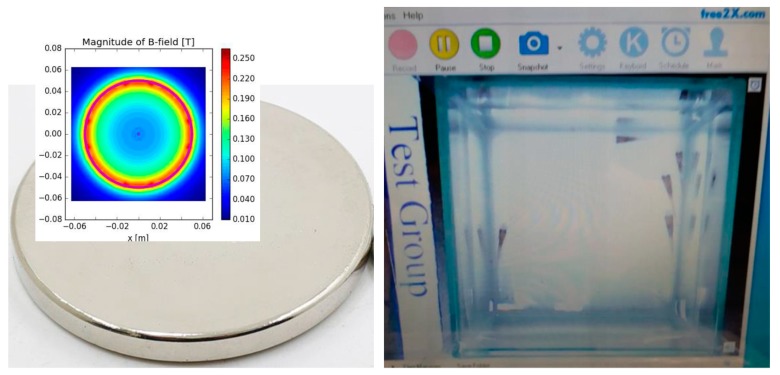
Illustration of the magnet and the tank keeping adult medaka fish recorded by the webcam.

**Figure 6 ijerph-16-00844-f006:**
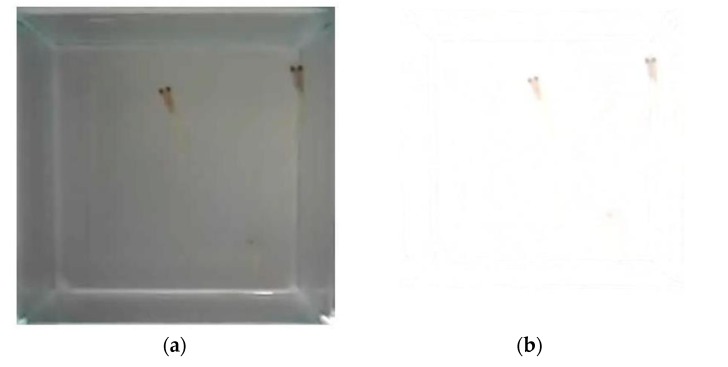
(**a**) Video before background subtraction; (**b**) Result after background subtraction with negative effect; (**c**) Video before image segmentation; (**d**) Result after image segmentation.

**Figure 7 ijerph-16-00844-f007:**
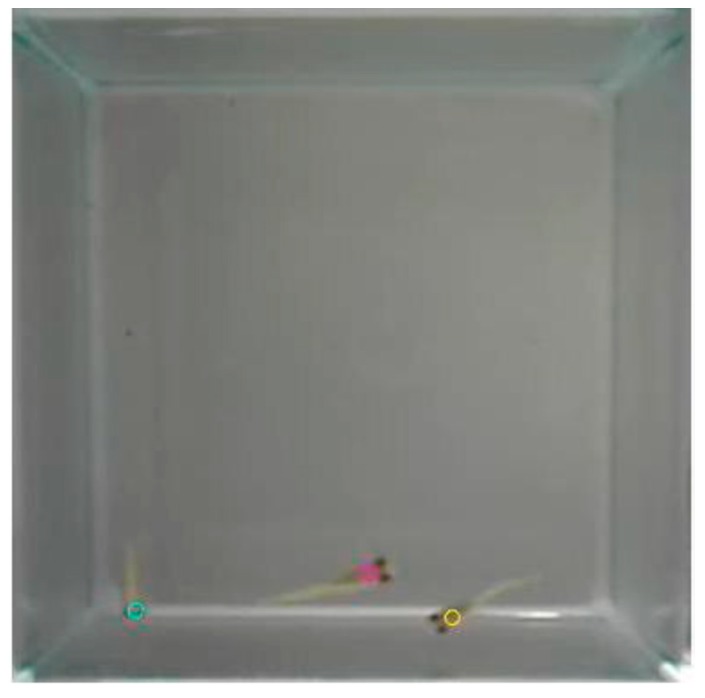
The capture of the new video with marked “O” on the coordinates.

**Figure 8 ijerph-16-00844-f008:**
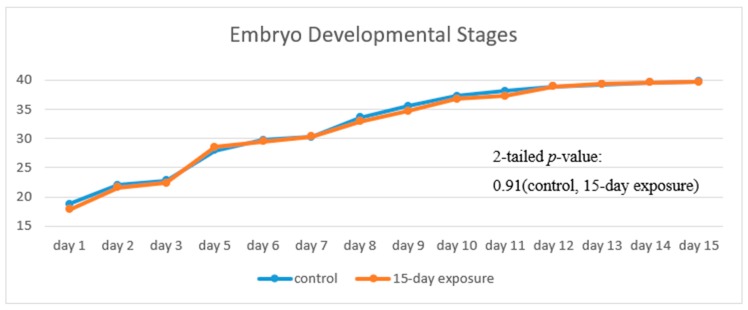
Stages of embryos development of the treated group with 15 days exposure in comparison with the typical stages of the control groups.

**Figure 9 ijerph-16-00844-f009:**
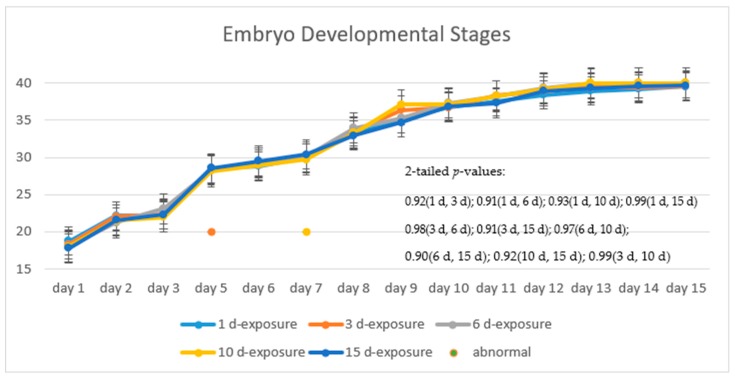
Stages of embryo development of the treated groups with different days of exposure.

**Figure 10 ijerph-16-00844-f010:**
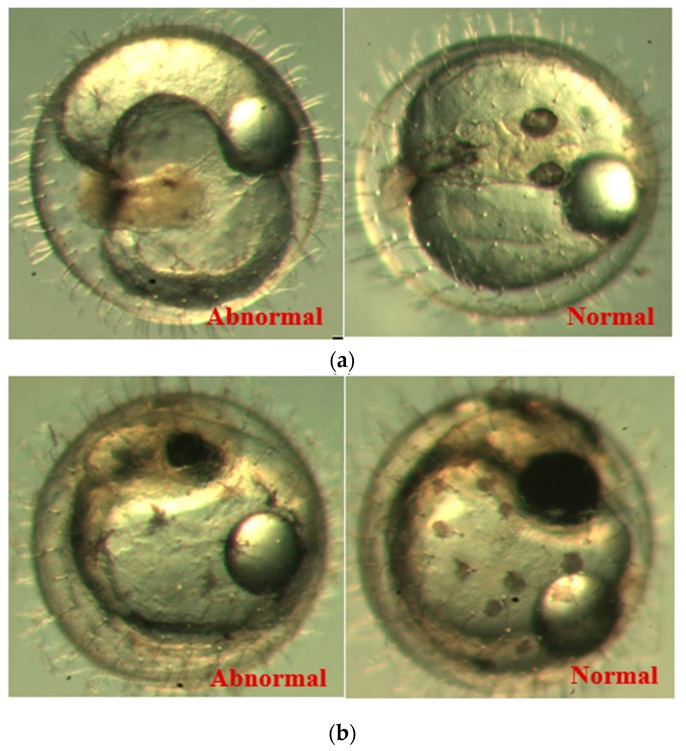
Abnormalities: (**a**) One of dish 5—abnormal body development; (**b**) One of dish 2—abnormal eye development.

**Figure 11 ijerph-16-00844-f011:**
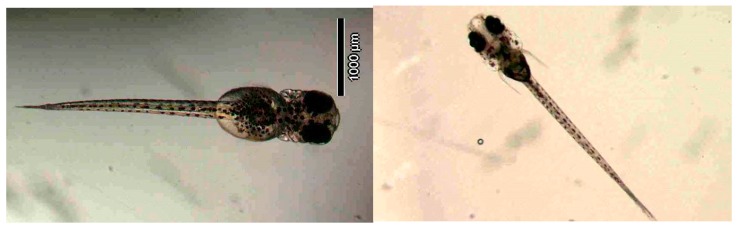
Illustration of medaka larvae 3 days post-hatching (**left**); and 13 days post-hatching (**right**).

**Figure 12 ijerph-16-00844-f012:**
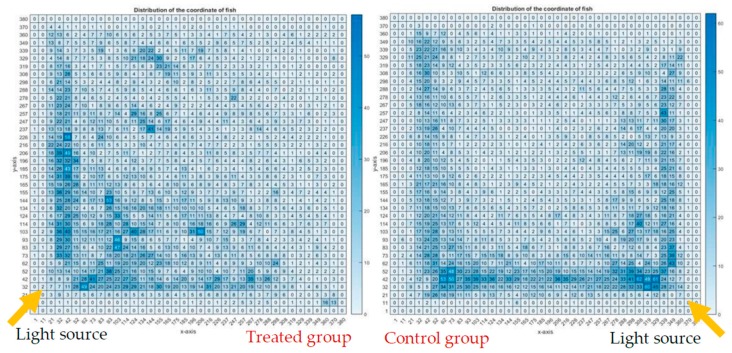
Heatmap charts in pixel format showing the fish being observed in a particular pixel in terms of the coordination of the fish.

**Figure 13 ijerph-16-00844-f013:**
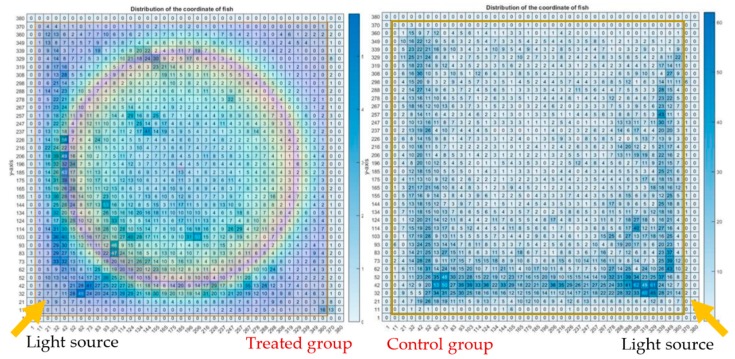
Heatmap chart of the coordinates of treated group with the magnet.

**Table 1 ijerph-16-00844-t001:** Time needed to hatch.

Embryos	Dish							
	1	2	3	4	5	6	7	8
Total hatched	15	14	15	14	14	15	15	13
Hatchability %	100	93.3	100	93.3	100	100	100	86.7
Average days need for hatching	16.8	12.7	12.7	17.1	15.7	13.7	16	14.6

**Table 2 ijerph-16-00844-t002:** Developmental stages of control groups and whole period exposure group (photos selected the embryo showing the typical stage [26] in the dish).

Stage	Standard	Control	Exposure
(a) 14 (Day 0) Pre-mid-gastrula stage	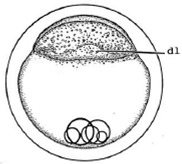	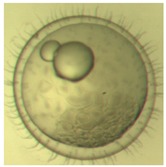	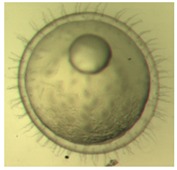
(b) 18 (Day 1) Late neurula stage (optic bud formation)	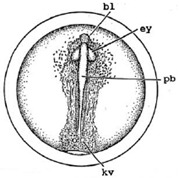	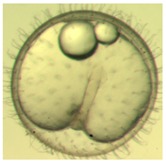	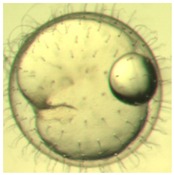
(c) 22 (Day 2) 9 somite stage (appearance of heart anlage)	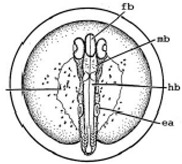	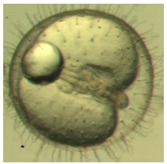	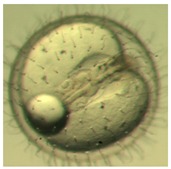
(d) 23 (Day 3) 12 somite stage (formation of tubular heart)	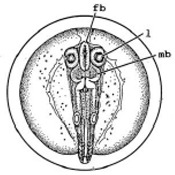	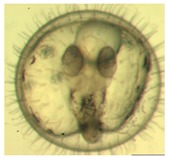	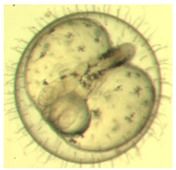
(e) 28 (Day 5) 30 somite stage (onset of retinal pigmentation)	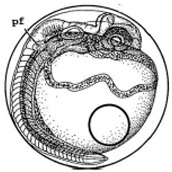	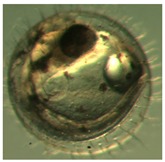	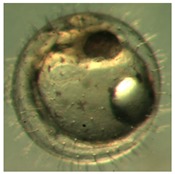
(f) 29 (Day 6) 34 somite stage (internal ear formation)	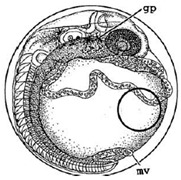	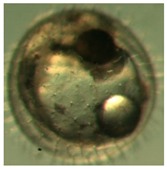	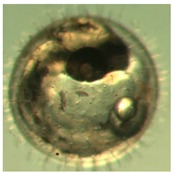
(g) 30 (Day 7) 35 somite stage (blood vessel development)	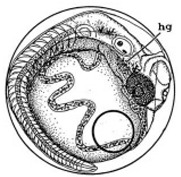	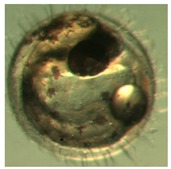	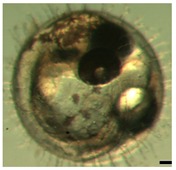
(h) 33 (Day 8) Stage at which notochord vacuolization is completed	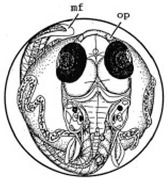	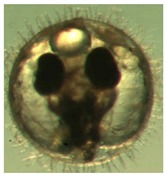	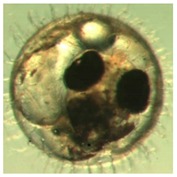
(i) 35 (Day 9) Stage at which visceral blood vessels form	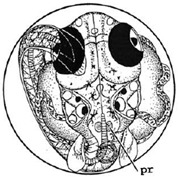	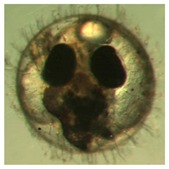	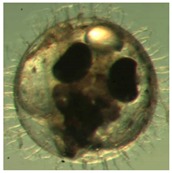
(j) 37 (Day 10) Pericardial cavity formation stage	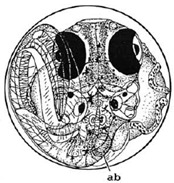	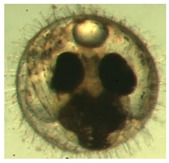	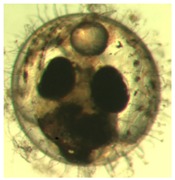
(k) 39 (Day 13) Hatching stage	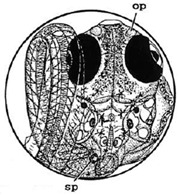	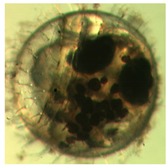	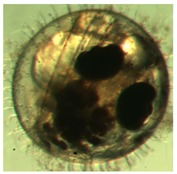
(l) 40 (Day 15) First fry stage	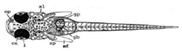	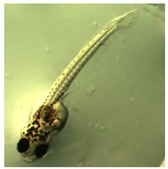	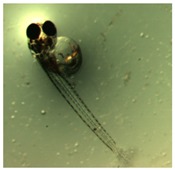

**Table 3 ijerph-16-00844-t003:** Average velocities of fish of the treated group and the control group.

Video	Control Group (pixel/frame)	Treated Group (pixel/frame)
1	1.6626	0.9027
2	1.8257	1.0455
3	2.0972	1.1478
4	2.1778	1.2754
5	2.3986	1.7030
6	2.5000	1.9220
7	2.5371	1.9970
8	2.6378	2.2628
9	3.0467	2.5263
10	3.2198	2.5601
11	3.2270	2.9128
12	3.6779	3.1247
Average	2.5840	1.9483

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
