# Peer review of "In Vivo Analysis of Embryo Development and Behavioral Response of Medaka Fish under Static Magnetic Field Exposures"

_ijerph, 2019, doi:10.3390/ijerph16050844_

Round 1
Reviewer 1 Report
Please refer to the attachment

Author Response
Dear Reviewer,
Thank you for your valuable advice and thoughtful comments, and pointing out the questions. We have reviewed and studied the comments carefully, and have tried our best to revise the manuscript; the revised manuscript according to the comments has been uploaded. The attached document is the responses to your comments to explain point-by-point further in details of the revision in the manuscript.
Thank you for your careful work.

Reviewer 2 Report
The author investigated the effects of static magnetic fields on embryo development and behavioral response of medaka fish and found that the swimming movement of the medaka fish was affected by 100 mT SMF but not the embryo development. This paper provided some very interesting and useful information to the field. However, there are still some points that should be addressed before it is considered for publication.
1. In the introduction part, there are numerous researches on the biological effects of static magnetic fields in vitro and in vivo. The authors should definitely do more literature search and cite more references. There are only 14 references, which is way too few.
2. The description of experimental methods is confusing. The detailed information about the magnets used should be provided, including the measured intensity of the magnetic field applied to the embryos in the dishes. Which magnetic pole was used in the behavioral response test? The authors should also add statistical analysis in the Materials and Methods.
3. The formula (2) should be explained carefully.
4. In line 153-155, the authors mentioned “The small abnormal rate can be considered as an occasional event such as accidental bacterial infection or developmental failure within ordinary probability”. There should be some references here.
5. In line 194-195, the authors thought that the “fishes of the treated group were likely to swim at the center, and otherwise for the control group”. But in figure 12 and figure 13, I suppose that the fish in the treated group were likely to swim to light source. Please double check this point.
6. The sentence in line 195-196 is not precise, medaka fish may simply prefer to stay in this weak SMF described in the experiment, because medaka fish may keep away high SMF or magnet with different poles, it is important to clearly describe the experimental conditions.
7. The English of the manuscript should be improved.
Author Response

(The authors gave the same response as above.)

Round 2
Reviewer 1 Report
I've looked over that paper and the authors have (mostly) satisfactorily addressed the points I raised, to the point that I'd agree for it to be published at this point.